# Incidence of long-term post-acute sequelae of SARS-CoV-2 infection related to pain and other symptoms: A systematic review and meta-analysis

Hiroshi Hoshijima[1], Takahiro Mihara[2], Hiroyuki Seki[3], Shunsuke Hyuga[4], Norifumi Kuratani[5], Toshiya Shiga[6]*

1 Division of Dento-oral Anesthesiology, Tohoku University Graduate School of Dentistry, Sendai, Miyagi, Japan, 2 Department of Anesthesiology and Critical Care Medicine, Yokohama City University Graduate School of Medicine, Yokohama, Kanagawa, Japan, 3 Department of Anesthesiology, Kyorin University School of Medicine, Mitaka, Tokyo, Japan, 4 Department of Anesthesiology, Kitasato University School of Medicine, Sagamihara, Kanagawa, Japan, 5 Department of Anesthesiology, Saitama Children's Medical Center, Saitama, Saitama, Japan, 6 Department of Anesthesiology and Pain Medicine, International University of Health and Welfare Ichikawa Hospital, Ichikawa, Chiba, Japan

* qzx02115@nifty.com

**Data Availability Statement:** All relevant data are within the article and its Supporting Information files.

## Abstract

### Background

Persistent symptoms are reported in patients who survive the initial stage of COVID-19, often referred to as "long COVID" or "post-acute sequelae of SARS-CoV-2 infection" (PASC); however, evidence on their incidence is still lacking, and symptoms relevant to pain are yet to be assessed.

### Methods

A literature search was performed using the electronic databases PubMed, EMBASE, Scopus, and CHINAL and preprint servers MedRχiv and BioRχiv through January 15, 2021. The primary outcome was pain-related symptoms such as headache or myalgia. Secondary outcomes were symptoms relevant to pain (depression or muscle weakness) and symptoms frequently reported (anosmia and dyspnea). Incidence rates of symptoms were pooled using inverse variance methods with a DerSimonian-Laird random-effects model. The source of heterogeneity was explored using meta-regression, with follow-up period, age and sex as covariates.

### Results

In total, 38 studies including 19,460 patients were eligible. Eight pain-related symptoms and 26 other symptoms were identified. The highest pooled incidence among pain-related symptoms was chest pain (17%, 95% confidence interval [CI], 11%-24%), followed by headache (16%, 95% CI, 9%-27%), arthralgia (13%, 95% CI, 7%-24%), neuralgia (12%, 95% CI, 3%-38%) and abdominal pain (11%, 95% CI, 7%-16%). The highest pooled incidence among other symptoms was fatigue (44%, 95% CI, 32%-57%), followed by insomnia (27%, 95% CI,

**Funding:** This study was supported by International University of Health and Welfare with funding for TS. The funders had no role in study design, data collection and analysis, decision to publish, or preparation of the manuscript.

**Competing interests:** The authors have declared that no competing interests exist.

10%-55%), dyspnea (26%, 95% CI, 17%-38%), weakness (25%, 95% CI, 8%-56%) and anosmia (19%, 95% CI, 13%-27%). Substantial heterogeneity was identified ($I^2$, 50–100%). Meta-regression analyses partially accounted for the source of heterogeneity, and yet, 53% of the symptoms remained unexplained.

## Conclusions

The current meta-analysis may provide a complete picture of incidence in PASC. It remains unclear, however, whether post-COVID symptoms progress or regress over time or to what extent PASC are associated with age or sex.

## Introduction

A broad range of symptoms have been reported to persist beyond the acute phase of SARS-CoV-2 virus infection [1–6]. These are referred to as "long COVID" [1, 3, 5, 6], "long-hauler" [5] or "Post-COVID-19 syndrome" [4, 5]. The National Institute of Health currently advocates calling these symptoms post-acute sequelae of SARS-CoV-2 infection (PASC) [7]. This syndrome is sometimes covered sensationally by news media or social networks, but little is known about its etiology, natural history, risk factors or therapeutic interventions. Even more, evidence on its incidence is still lacking.

On a cellular level, the spike protein in the SARS-CoV-2 virus combines with angiotensin-converting enzyme 2 (ACE2) receptor, invades human cells, and injures multiple organs [8]. Central and peripheral nerve systems are one of the most susceptible targets for SARS-CoV-2 virus (neurotropism) [9]. Frequently reported symptoms range from fatigue, muscle weakness and memory loss to anosmia, ageusia, confusion and headache [1–6, 10]. Some of these symptoms are directly or indirectly related to chronic pain, often worsening quality of life for a long period. As well, a prolonged period of mechanical ventilation in the ICU may cause what is called "post intensive care syndrome" or "ICU-acquired weakness" [9], manifesting as cognitive dysfunction, muscle atrophy, sensory disruption and joint-related pain [8]. These patients will be at elevated risk of developing chronic pain. Furthermore, SARS-CoV-2 virus causes "cytokine storm", which aggravates damage in multiple tissues including joints and muscles that possibly triggers pain-related symptoms [8]. A recent study [11] has shown that the prevalence of new-onset headache was substantially higher in COVID-19 survivors compared with those in controlled subjects. Nevertheless, pain in COVID-19 survivors has been underestimated or paid little attention. Treatment of pain in such patients is prone to be of low priority, especially due to overburdened healthcare services or difficulty in consulting with a specialist over the course of the pandemic [12].

As pain clinicians, we believe that understanding and managing pain-related symptoms along with other symptoms will help to improve the quality of life of SARS-CoV-2 survivors. Therefore, we collected currently available evidence and conducted a rapid systematic review and meta-analysis of observational studies to determine the incidence of pain-related and other symptoms in SARS-CoV-2 convalescents.

## Methods

We defined long-term complications as symptoms from which patients suffered for more than 1 month after onset of the first COVID-19 symptoms or after discharge from hospital. A meta-

analysis was conducted according to the reporting guidelines for the Preferred Reporting Items for Systematic Reviews and Meta-Analyses (PRISMA) [13]. The protocol was previously registered on PROSPERO (CRD42021228393).

## Search strategy

Three reviewers (HH, SH and TS) searched the electronic databases PubMed, EMBASE, Scopus and CHINAL and preprint servers MedRχiv and BioRχiv. No language restriction was applied. The last search was done on January 15, 2021. The full search strategy is described in S1 Appendix. Reference lists of all identified articles on "long-covid" were manually searched. All relevant references obtained in the RIS (Research Information Systems) formats were transferred to EndNote X8.2 (Clarivate, Philadelphia, PA, USA) and web-platform manager Covidence (Melbourne, Australia).

## Eligibility criteria

Studies involving adults (>18 years old) with a confirmed diagnosis of SARS-CoV-2 were included, as were studies that followed up patients for a minimum of 2 weeks after discharge. Studies only focusing on acute symptoms from admission without any mention of long-term symptoms were excluded. Prospective and retrospective cohort studies were also included. Reviews, editorials, meta-analyses, case reports, case series and case-control studies were excluded. Regardless of whether a reported symptom was pain-related or not, studies reporting any relevant "long-covid" symptoms were included. Studies reporting only radiological findings of lung or brain were excluded.

## Screening and data extraction

Two reviewers (HH and TS) independently screened titles and abstracts of the obtained references by using Covidence. Disagreements were resolved by discussion with a third reviewer (SH). Data extraction was performed by five reviewers (HH, TM, HS, SH and TS), and the extracted data were saved in an Excel spreadsheet. Extracted data included study setting, country where study was performed, patient setting, diagnostic criteria of SARS-CoV-2, respiratory support, mean age, percentage of males, follow-up period and information for evaluating study quality. The primary outcome was defined as pain-related symptoms such as headache or myalgia. The secondary outcome was defined as symptoms other than but relevant to pain such as depression or fatigue, or frequently reported symptoms such as anosmia or dyspnea. When data were reported as a graph only, we reproduced numerical data using Plot Digitizer (http://plotdigitizer.sourceforge.net).

## Assessment of study quality

The Newcastle-Ottawa scale for cohort studies [14] was used to assess the methodological quality of the studies by the five reviewers. Briefly, the scale consists of three subcategories: selection, comparability and outcome and 9 items. However, we focused on pooled incidence of long-COVID symptoms rather than any treatment effects and all patients exposed to SARS-Cov-2 virus (excluding the non-exposed cohort); therefore, some of the items were impossible to evaluate such as selection of the non-exposed cohort and comparability. Thus, these two items were excluded from the checklist, and study quality was assessed by the rest of the items (S1 Appendix). One point was given for each item, for a maximum score of 6 and a minimum score of 0.

## Statistical analysis

At least 3 studies were required per one symptom, due to constraints in performing data synthesis. The proportions of symptoms in an individual study were pooled using inverse variance methods following logit transformation [15]. Between-study variances were quantified using the DerSimonian-Laird estimator [16]. To calculate 95% confidence intervals (CIs) in an individual study, the Clopper-Pearson interval was used. The $I^2$ statistic was used as a measure of heterogeneity ($I^2$ >60%: high heterogeneity; 40–60%: moderate heterogeneity; <40%: low heterogeneity). Sensitivity analysis and subgroup analysis were not performed because our aim in this meta-analysis was to exploratorily collect currently available evidence of overall incidence. Because of possible selection bias, the top 3 most frequent symptoms were ranked in each study, and we aggregate them in an overall ranking.

We explored the source of heterogeneity by meta-regression using a mixed-effects model [17]. We incorporated three covariates (follow-up period, mean age and percentage of males) with fixed effects, and each study as a random effect. $R^2$ was used as a measure of the amount of heterogeneity that could be accounted for by the covariate. Briefly, an index $R^2$ value is defined as the ratio of explained heterogeneity to total heterogeneity, with a range of 0% to 100%. We plotted the logit transformed incidence of each symptom on the Y axis and the covariate on the X axis, along with predicted regression line (bubble plot).

Statistical significance was set at a 2-tailed $\alpha$ = .05. To evaluate small-study effects (publication bias), a funnel plot was depicted and Egger test was performed [18], with significance applied at $P < .010$. All statistical analyses were conducted using the *meta* package of R version 4.0.3 (The R Foundation for Statistical Computing) and RStudio 1.4 (Boston, MA).

## Results

The initial search yielded 1290 citations, of which 105 potentially relevant studies were assessed in full text. Thirty-five studies [19–53] were included. Three studies were manually added according to a reviewer's suggestion during the revision process of the manuscript [54–56]. Finally, 38 studies comprising 19,460 patients were included in the meta-analysis (Fig 1).

All studies were written in English. A summary of the included studies is presented in S1 Table. Studies were reported mainly from Europe, followed by the USA and China. Follow-up duration ranged from 0.5 to 7 months.

The results of the Newcastle-Ottawa scale are shown in S1 Table. Most of the studies (33/38, 87%) scored 5 or 6, and the median score of the 38 studies was 5 (range: 3–6).

The results of each symptom on the forest plot are shown in S1 Fig. The pooled incidence of each primary and secondary outcome is shown in order of frequency in Figs 2 and 3, respectively.

The most frequent symptom among pain-related symptoms was chest pain (17%, 95% CI, 12%-25%), followed by headache (16%, 95% CI, 9%-27%), arthralgia (13%, 95% CI, 7%-24%), neuralgia (12%, 95% CI, 3%-38%) and abdominal pain (11%, 95% CI, 7%-16%). The most frequent symptom in the secondary outcomes was fatigue (45%, 95% CI, 32%-59%), followed by insomnia (26%, 95% CI, 9%-57%), dyspnea (25%, 95% CI, 15%-38%), weakness (25%, 95% CI, 8%-56%) and anosmia (19%, 95% CI, 13%-27%).

Regarding the most frequent symptoms reported in each article, 35 articles were included after excluding articles focusing on a single symptom such as headache [39]. Our results showed that the three most frequent symptoms summarized were fatigue (n = 20), dyspnea (n = 17), cough (n = 13), followed by anosmia (n = 12) and fever (n = 6) (S1 Table).

The results of $R^2$ obtained by meta-regression are shown in the Table 1, and those of the statistical analyses and bubble plots are detailed in S1 Fig.

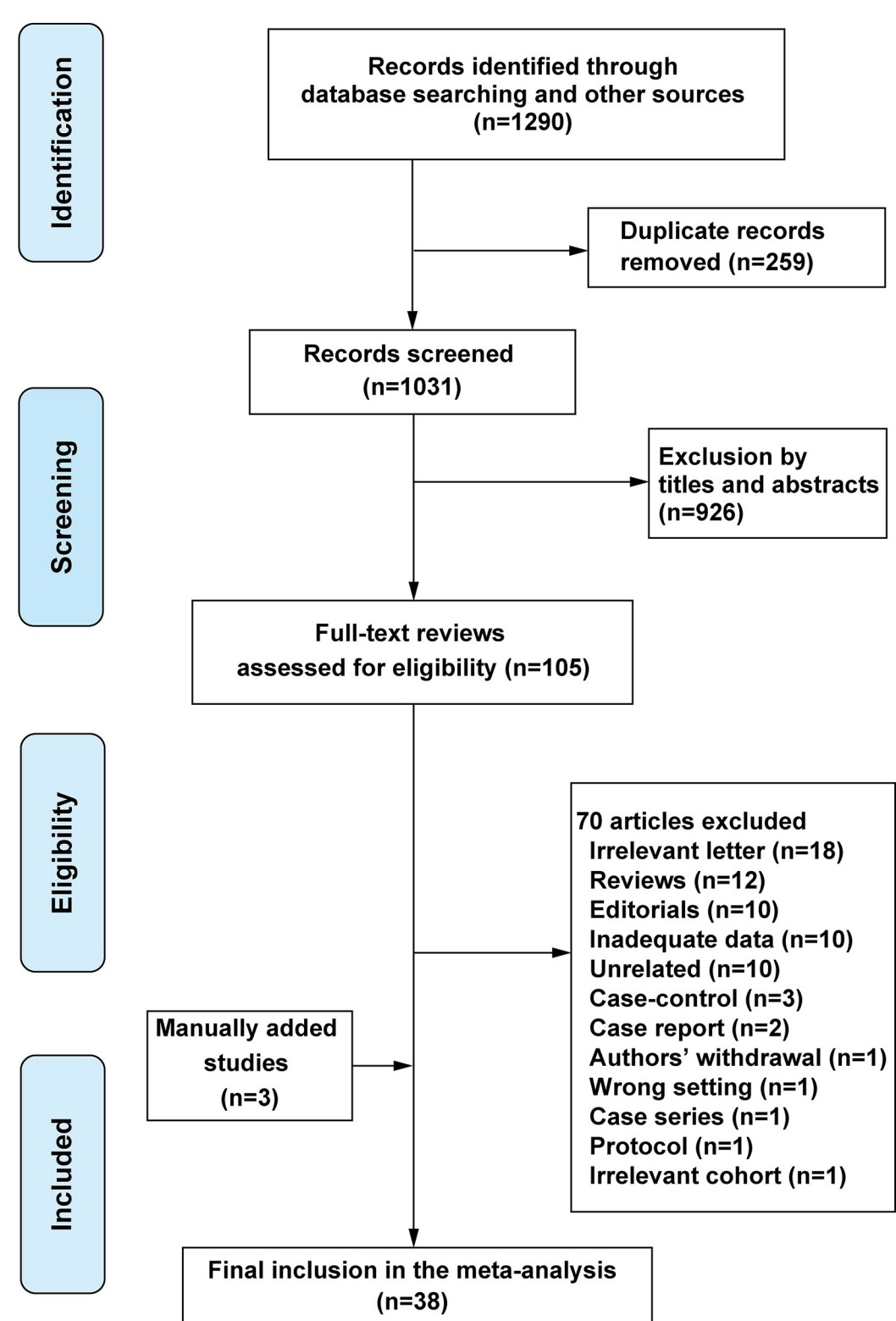

**Fig 1. PRISMA flow diagram for literature search, study screening and selection.**

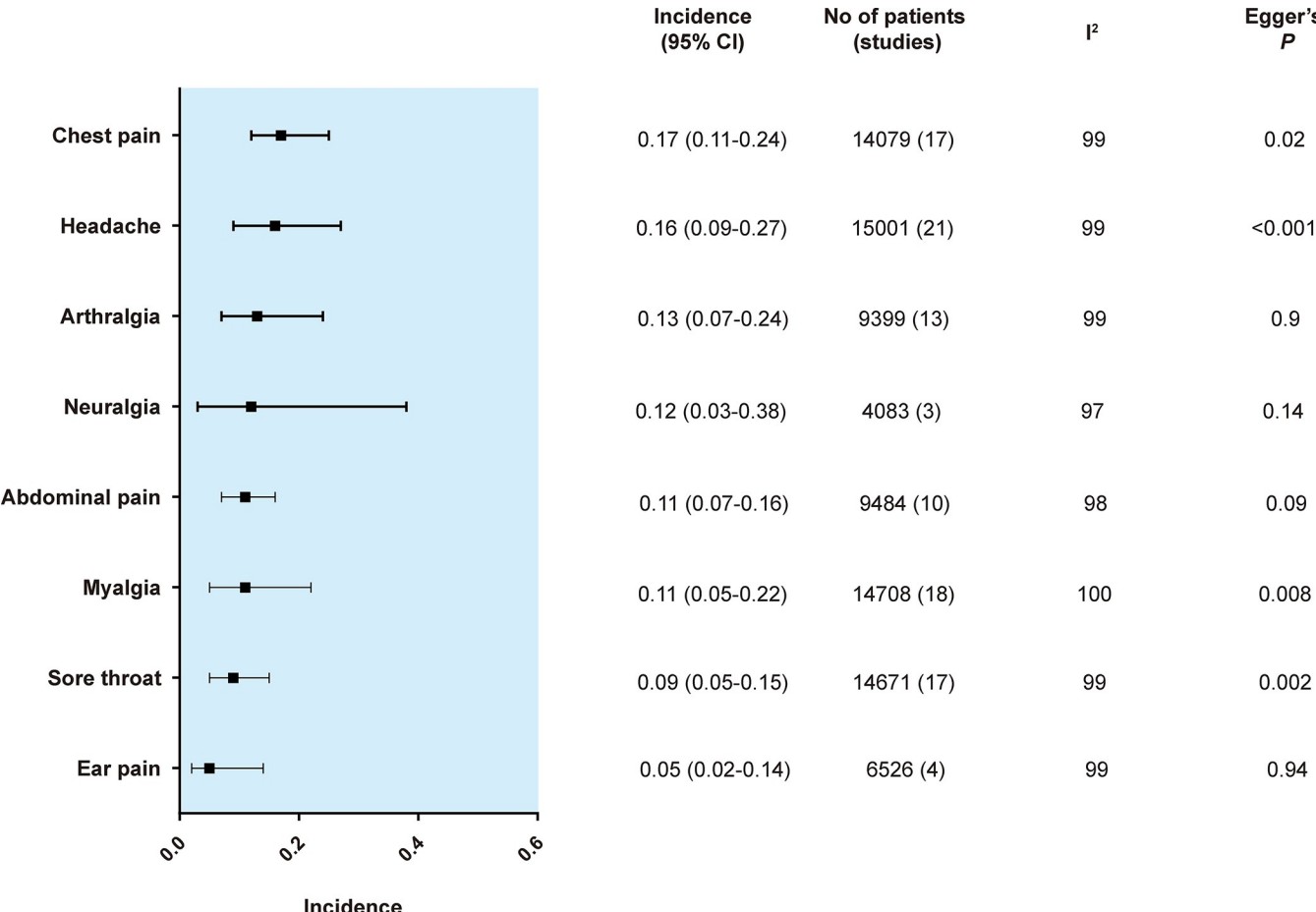

**Fig 2. Summary random effects estimates with 95% confidence interval (CI) from 8 meta-analyses on the incidence of pain-related symptoms.** $I^2$ represents the degree of heterogeneity, and Egger's P represents publication bias.

Among pain-related symptoms, significant correlations were identified only for neuralgia: however, only three studies with this symptom were included. For instance, the regression coefficient for follow-up period was 0.39 (logit transformed), which means that every one month of follow-up corresponds to an increase of 1.45 units (45% increase) in prevalence in patients who developed neuralgia after acute COVID-19 infection. For the other symptoms, significant correlations were found for insomnia, dyspnea, weakness, anosmia, cough, ageusia, memory impairment, depression, anxiety, nasal blockage, weight loss, sputum, chills and nausea. Among the symptoms overall, 53% remained unexplained when using the three covariates in the model.

The results of the funnel plots are shown in S1 Fig. For pain-related symptoms, small-study effects as assessed by Egger test were observed for 4 of 8 symptoms. For other symptoms, small-study effects were observed for 15 of 26 symptoms. In total, small-study effects were identified for 56% of the symptoms.

## Discussion

The current meta-analysis suggested three main findings. First, pain-related symptoms in COVID-19 survivors were multifarious with an incidence of 5–17%. Second, other symptoms were more multifaceted with incidences ranging from 2% to 45%. Third, every symptom

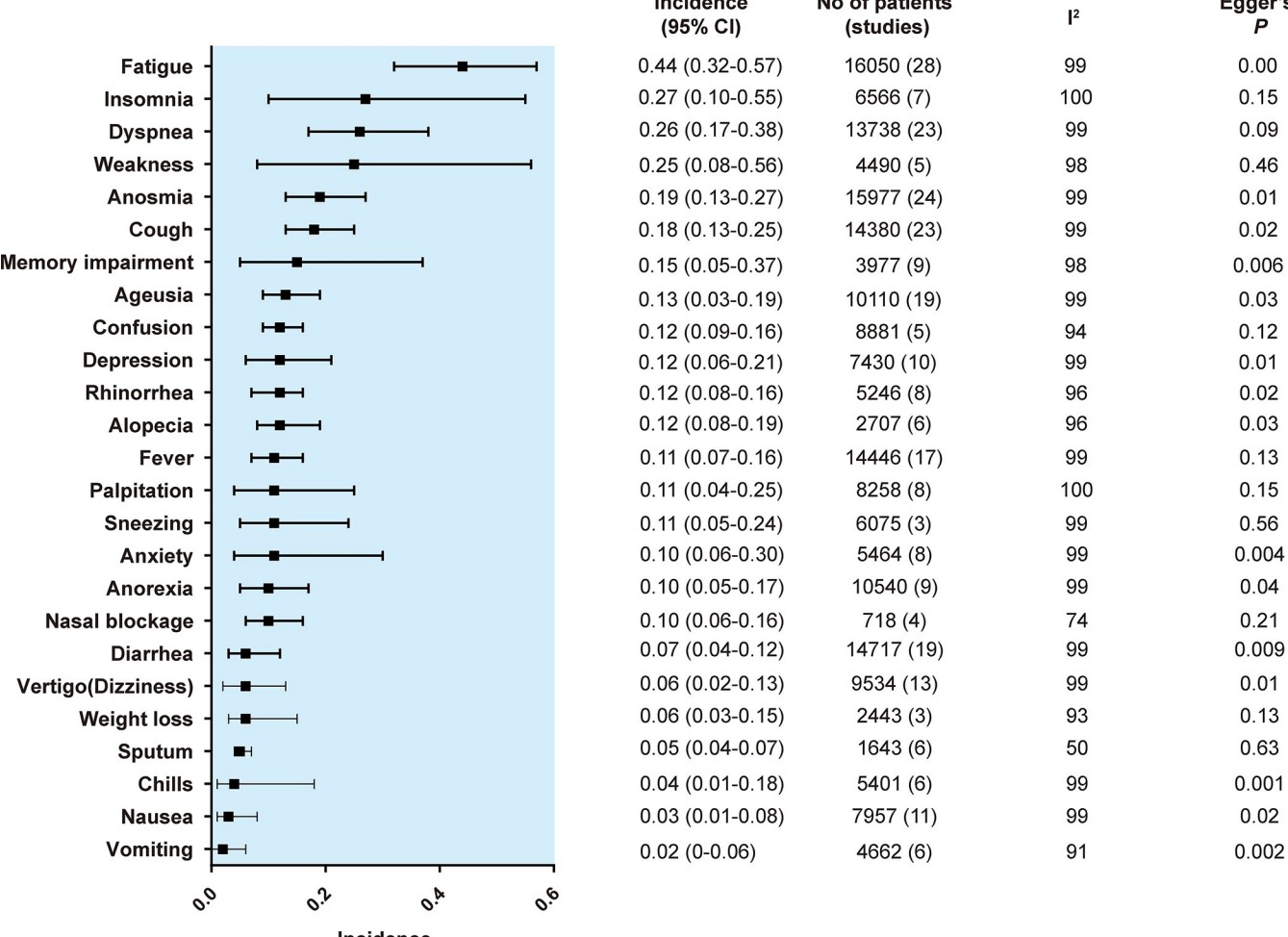

**Fig 3. Summary random effects estimates with 95% confidence interval (CI) from 8 meta-analyses on the incidence of other symptoms.** $I^2$ represents the degree of heterogeneity, and Egger's P represents publication bias.

varied extensively in its incidence, and the three major covariates (follow-up, age and sex) could not explain the heterogeneity.

Among pain-related symptoms, the highest pooled incidence was chest pain (17%), followed by headache (16%), arthralgia (13%), neuralgia (12%) and abdominal pain (11%). Chest pain is also referred to as "lung burn", which is considered to be a result of lung injury by SARS-CoV-2 infection [6]. Alternatively, other researchers pointed out that chest pain may result from pericarditis caused by infection [29]. Headache is one of the most common central nervous system symptoms in patients with SARS-CoV-2 infection [57, 58]. It can persist over the period of the initial infection [59], or it can develop as a new-onset form during healing [11]. Proposed mechanisms include direct invasion of trigeminal nerve endings by SARS-CoV-2 via disruption of the brain-blood barrier, trigeminovascular activation via involvement of endothelial cells with ACE2 expression, or triggering of perivascular trigeminal nerve endings by release of cytokines and pro-inflammatory mediators [59].

Among other symptoms, almost half of the patients developed fatigue. Generally, fatigue is considered to be closely related to chronic pain. Myalgic encephalomyelitis/chronic fatigue syndrome (ME/CFS) [60] or fibromyalgia [61] are good examples. A recent report suggested

**Table 1. Results of meta-regression to explore the source of heterogeneity.**

| | No of studies | Heterogeneity ($I^2$) | Amount of heterogeneity accounted for ($R^2$) (%) | | |
|---|---|---|---|---|---|
| | | | Follow-up period | Age | Gender (male) |
| **Pain-related symptoms** | | | | | |
| Chest pain | 17 | 99 | 0 | 0 | 0 |
| Headache | 21 | 99 | 0 | 0 | 0 |
| Arthralgia | 13 | 99 | 0 | 0 | 0 |
| Neuralgia | 3 | 97 | **100 (+)** | **92 (-)** | **69 (+)** |
| Abdominal pain | 10 | 98 | 0 | 0 | 0 |
| Myalgia | 18 | 100 | 0 | 0 | 0 |
| Sore throat | 17 | 99 | 0 | 0 | 0 |
| Ear pain | 4 | 99 | 0 | 0 | 0 |
| **Other symptoms** | | | | | |
| Fatigue | 28 | 99 | 0 | 0 | 0 |
| Insomnia | 7 | 100 | **44 (+)** | **28 (-)** | **75 (+)** |
| Dyspnea | 23 | 99 | **34 (+)** | 0 | 0 |
| Weakness | 5 | 98 | 0 | **64 (-)** | 0 |
| Anosmia | 24 | 99 | 0 | **35 (-)** | 0 |
| Cough | 23 | 99 | 0 | **42 (-)** | **10 (-)** |
| Ageusia | 19 | 99 | 0 | 0 | **19 (-)** |
| Memory impairment | 9 | 98 | **73 (+)** | **5 (+)** | **40 (+)** |
| Confusion | 5 | 94 | 0 | 0 | 0 |
| Depression | 10 | 99 | **55 (-)** | 0 | **23 (+)** |
| Fever | 17 | 99 | 0 | 0 | 0 |
| Rhinorrhea | 8 | 98 | 0 | 0 | 0 |
| Anxiety | 8 | 99 | **66 (+)** | 0 | **67 (+)** |
| Palpitation | 8 | 100 | 0 | 0 | 0 |
| Sneezing | 3 | 99 | 0 | 0 | 0 |
| Alopecia | 6 | 96 | 0 | 0 | 0 |
| Anorexia | 9 | 99 | 0 | 0 | 0 |
| Nasal blockage | 4 | 74 | **33 (-)** | 0 | **70 (+)** |
| Diarrhea | 19 | 99 | 0 | 0 | 0 |
| Vertigo (Dizziness) | 13 | 99 | 0 | 0 | 0 |
| Weight loss | 13 | 93 | **23 (-)** | 0 | **40 (+)** |
| Sputum | 6 | 50 | **65 (-)** | 0 | 0 |
| Chills | 6 | 99 | **72 (+)** | 0 | **77 (+)** |
| Nausea | 11 | 99 | **54 (+)** | **1 (-)** | 0 |
| Vomiting | 6 | 91 | **7 (+)** | 0 | 0 |

$R^2$ represents a measure of the amount of heterogeneity that can be explained by the covariate. Bold numbers indicate that a significant correlation was found between the symptom and the covariate. + or–in parenthesis indicates a positive or negative coefficient in the regression model. Note that for insomnia and follow-up period, for instance, the incidence of insomnia is significantly higher when the follow-up period increases (positive correlation). Note that for ageusia and sex, the incidence of ageusia is significantly higher when the ratio of males in a study population decreases (inverse correlation).

that there are similarities and overlap in pathology between long COVID symptoms and ME/CFS [4, 60]. As fatigue is often refractory to a single approach, holistic management such as rehabilitation or cognitive behavioral therapy is required [6]. Weakness, often accompanied by myalgia and arthralgia, is a musculoskeletal manifestation of SARS-CoV-2 infection [62]. Muscle fiber atrophy, extensive use of corticosteroids, prolonged mechanical ventilation or systematic inflammation may be the causes of weakness [62].

From the results of the meta-regression, the incidence of neuralgia was significantly associated with follow-up period, age or sex to some extent; however, only 3 studies were included with this symptom. Therefore, it is difficult to consider this result to be valid. As another example, an inverse association was found between the incidence of weakness and age, but we could not explain this well. In any case, we are aware that these statistical models are preliminary and exploratory, and 53% of symptoms were not explainable despite three typical covariates being incorporated into the model. Symptoms of long COVID are reported to be on-and-off, cyclic or multiphasic [5], which is why the linear regression model did not fit well.

During the course of our study, a similar meta-analysis related with pain were published [63]. Although this might weaken the originality of our study, in the publication, the most frequently reported pain-related symptoms and their incidence were comparable with those reported in our studies. For instance, chest pain is the most reported symptoms, and its incidence ranges between 7.8–23.6%.

This study has several limitations. First, considerable heterogeneity was found in most of the symptoms, and meta-regression could not explain it in just over half of the symptoms. Possible reasons may be the following: in the light of the nature of observational studies, the subjects are not homogenous. The current study includes reports from a wide range of countries; thus, the definition and diagnostic criteria of symptoms might vary from study to study. The majority of data were collected via telephone interview or online survey. A face-to-face visit was not always possible during the COVID-19 pandemic, and therefore, recall bias might possibly have occurred. Second, pain-related symptoms were less frequent compared with other symptoms. Selection bias in each study might be possible because pain-related symptoms might be underdiagnosed. Third, the current study did not include "brain fog", "covid toe" or "post-exertional malaise", which are widely known as post-COVID symptoms [2, 6, 26, 62], because these symptoms did not fulfill our inclusion criteria of at least three studies being required for data synthesis. However, we will be able to update this review if more reports are published on these symptoms in the future. Fourth, publication bias was identified for 56% of all symptoms. This suggested that the point estimates of the incidence of symptoms in our study might have been overestimated or underestimated. Lastly, studies performed in different geographical regions might be a potential factor contributing to the heterogeneity, which was suggested in a recent meta-analysis [64].

## Conclusions

The present meta-analysis highlighted the incidence of pain-related and other typical symptoms in patients with PASC. It remains uncertain whether post-COVID symptoms progress or regress over time and to what extent PASC are associated with age or sex.

## Supporting information

**S1 Appendix. Literature search strategy, R code and the Newcastle-Ottawa quality assessment scale.**
(DOCX)

**S1 Table. Summary of studies, the results of the Newcastle-Ottawa quality assessment and most frequent symptoms ranked in each study.**
(DOCX)

**S1 Fig. Forest plot, bubble plot and funnel plot in each symptom.**
(DOCX)

## Author Contributions

**Conceptualization:** Hiroshi Hoshijima, Toshiya Shiga.

**Data curation:** Hiroshi Hoshijima, Takahiro Mihara, Hiroyuki Seki, Shunsuke Hyuga, Toshiya Shiga.

**Formal analysis:** Takahiro Mihara, Norifumi Kuratani, Toshiya Shiga.

**Investigation:** Hiroshi Hoshijima, Takahiro Mihara, Hiroyuki Seki, Shunsuke Hyuga, Norifumi Kuratani, Toshiya Shiga.

**Methodology:** Takahiro Mihara, Norifumi Kuratani, Toshiya Shiga.

**Project administration:** Toshiya Shiga.

**Resources:** Shunsuke Hyuga.

**Software:** Takahiro Mihara, Norifumi Kuratani, Toshiya Shiga.

**Supervision:** Norifumi Kuratani, Toshiya Shiga.

**Validation:** Hiroyuki Seki, Norifumi Kuratani.

**Visualization:** Takahiro Mihara, Toshiya Shiga.

**Writing – original draft:** Toshiya Shiga.

**Writing – review & editing:** Hiroshi Hoshijima, Takahiro Mihara, Hiroyuki Seki, Shunsuke Hyuga, Norifumi Kuratani.

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
