## [Decision Letter · Decision Letter 0]

9 Nov 2021

PONE-D-21-13798

Incidence of Long-term Post-acute Sequelae of SARS-CoV-2 Infection Related to Pain and Other Symptoms: A Living Systematic Review and Meta-analysis

PLOS ONE

Dear Dr. Shiga,

Thank you for submitting your manuscript to PLOS ONE. After careful consideration, we feel that it has merit but does not fully meet PLOS ONE’s publication criteria as it currently stands. Therefore, we invite you to submit a revised version of the manuscript that addresses the points raised during the review process.

Please address the issues and revise accordingly.

We look forward to receiving your revised manuscript.

Kind regards,

Academic Editor

PLOS ONE

Journal Requirements:

2. Thank you for stating the following financial disclosure: "NO"

3. Thank you for stating the following in your Competing Interests section: "NO"

5. Please remove all personal information, ensure that the data shared are in accordance with participant consent, and re-upload a fully anonymized data set. 

Reviewers' comments:

Reviewer's Responses to Questions

**Comments to the Author**

1. Is the manuscript technically sound, and do the data support the conclusions?

Reviewer #1: Yes

Reviewer #2: Yes

2. Has the statistical analysis been performed appropriately and rigorously? 

Reviewer #1: Yes

Reviewer #2: I Don't Know

3. Have the authors made all data underlying the findings in their manuscript fully available?

Reviewer #1: Yes

Reviewer #2: Yes

4. Is the manuscript presented in an intelligible fashion and written in standard English?

Reviewer #1: Yes

Reviewer #2: Yes

5. Review Comments to the Author

Reviewer #1: I read with interest about this comprehensive systemic review of post-acute COVID-19 symptoms. Generally, the article was well written with adequate meta-analyses. I still had some concerns about this study.

1. At least some important studies were not included in the current study. I suggest authors include these studies and re-analyze your results.

a. Chopra, V., Flanders, S. A. & O’Malley, M. Sixty-day outcomes among patients hospitalized with COVID-19. Ann. Intern. Med. (2020).

b. Garrigues, E. et al. Post-discharge persistent symptoms and health-related quality of life after hospitalization for COVID-19. J. Infect. 81, e4–e6 (2020).

2. The last literature research was performed on January 15, 2021. The post-acute COVID-19 symptoms had been reported more in detail after that time. Since this study is aimed to be a living meta-analysis, is it feasible for authors to include newer publised studies after Jan 2021. At least the following should be considered:

a. Huang L, Yao Q, Gu X, et al. 1-year outcomes in hospital survivors with COVID-19: a longitudinal cohort study. Lancet. 2021; 398(10302):747-758. →Updated results from reference 31.

b. Moreno-Pérez, O. et al. Post-acute COVID-19 syndrome. Incidence and risk factors: a Mediterranean cohort study. J. Infect. (2021)

Reviewer #2: The authors use a living systematic review and meta-analysis to depict the incidence of long-term post-acute sequelae of SARS-CoV-2 infection related to pain and other symptoms. They defined long-term complications as symptoms from which patients suffered for more than one month after onset of the first COVID-19 symptoms or after discharge from hospital. There are several concerns as follows:

Major concern 1: From the results, pain-related symptoms are much less frequently noticed compared to the secondary outcome, symptoms relevant to pain such as fatigue, insomnia, dyspnea, weakness. I am wondering whether the clinical setting could be a source of selection bias. For example, a patient underwent invasive treatment might be highly sedative. The pain would be masked initially, and its related symptoms may flair up in later disease courses.

Major concern 2: Theoretically, all symptoms should be thoroughly observed from estimated onset of the disease. It would be better to have a consistent format for checking up each symptom with daily bases. Since the available published data bear witness to a high heterogeneity, I suggest the authors make a summary table to show the common symptoms according to individual report.

Major concern 3: The authors smartly described “post intensive care syndrome” or “ICU-acquired weakness" in Discussion Section. However, the sensitivity analyses based on ICU and Non-ICU patients are lacking in present draft. If the authors can provide this assessment, it would be also helpful to clarify the Major concern 1.

Minor concern, Line 216: COVID-19 survivors were multifarious with "an" incidence of 5-17%.

6. PLOS authors have the option to publish the peer review history of their article (what does this mean?). If published, this will include your full peer review and any attached files.

Reviewer #1: **Yes: **Wei-Chih Chen

Reviewer #2: No

---

## [Author Response · Author response to Decision Letter 0]

20 Mar 2022

Answers to the reviewers’ comments.

Reviewer #1: I read with interest about this comprehensive systemic review of post-acute COVID-19 symptoms. Generally, the article was well written with adequate meta-analyses. I still had some concerns about this study.

Thank you for your comments.

First of all, we attempted to update the latest articles. After screening with the same search strategy, we found a skyrocketing number of articles on this theme: More than 2,000 articles from Scopus; around 20,000 from medRxiv, and more than 400 from EMBASE. After we discussed this, we considered it impossible, within this short period of revision time, to screen, extract data, re-analyze, and extensively rewrite the manuscript. Therefore, we decided to include the only studies you have pointed out, without altering the current search period. We kindly ask for your understanding in advance.

1. At least some important studies were not included in the current study. I suggest authors include these studies and re-analyze your results.

a. Chopra, V., Flanders, S. A. & O’Malley, M. Sixty-day outcomes among patients hospitalized with COVID-19. Ann. Intern. Med. (2020).

b. Garrigues, E. et al. Post-discharge persistent symptoms and health-related quality of life after hospitalization for COVID-19. J. Infect. 81, e4–e6 (2020).

We have completed re-analyses after including these studies.

2. The last literature research was performed on January 15, 2021. The post-acute COVID-19 symptoms had been reported more in detail after that time. Since this study is aimed to be a living meta-analysis, is it feasible for authors to include newer publised studies after Jan 2021. At least the following should be considered:

a. Huang L, Yao Q, Gu X, et al. 1-year outcomes in hospital survivors with COVID-19: a longitudinal cohort study. Lancet. 2021; 398(10302):747-758. →Updated results from reference 31.

b. Moreno-Pérez, O. et al. Post-acute COVID-19 syndrome. Incidence and risk factors: a Mediterranean cohort study. J. Infect. (2021)

We have completed re-analyses after including Moreno-Pérez. As you pointed out, the Huang et al. study is certainly an update of reference 31; however, we did not include this study because it was published far beyond the current search period.

Finally, we have included the three articles that you pointed out.

In addition, we excluded the term “Living” from the manuscript title as this is no longer a “living” meta-analysis.

 

Reviewer #2: 

The authors use a living systematic review and meta-analysis to depict the incidence of long-term post-acute sequelae of SARS-CoV-2 infection related to pain and other symptoms. They defined long-term complications as symptoms from which patients suffered for more than one month after onset of the first COVID-19 symptoms or after discharge from hospital.

Thank you for your time spent in reviewing our manuscript.

There are several concerns as follows:

Major concern 1: From the results, pain-related symptoms are much less frequently noticed compared to the secondary outcome, symptoms relevant to pain such as fatigue, insomnia, dyspnea, weakness. I am wondering whether the clinical setting could be a source of selection bias. For example, a patient underwent invasive treatment might be highly sedative. The pain would be masked initially, and its related symptoms may flair up in later disease courses.

We agree with you that pain-related symptoms are much less frequently noticed. We added this comment as a limitation in the limitations paragraph in the Discussion section. Closely related to your concern 3, we could not perform sensitivity analyses between ICU and non-ICU patients in this revision.

Major concern 2: Theoretically, all symptoms should be thoroughly observed from estimated onset of the disease. It would be better to have a consistent format for checking up each symptom with daily bases. Since the available published data bear witness to a high heterogeneity, I suggest the authors make a summary table to show the common symptoms according to individual report.

The top three most frequent symptoms were ranked in each study, and we summarized them in an overall ranking. Our results showed that most frequent symptoms summarized were fatigue (n=20), followed by dyspnea (17), cough (13), anosmia (12) and fever (6).

The detailed results are described in the Results section and eTable 3.

Major concern 3: The authors smartly described “post intensive care syndrome” or “ICU-acquired weakness" in Discussion Section. However, the sensitivity analyses based on ICU and Non-ICU patients are lacking in present draft. If the authors can provide this assessment, it would be also helpful to clarify the Major concern 1.

To perform a sensitivity analysis between ICU and non-ICU patients, we thoroughly reviewed all of the studies we included. However, we found that only two of the studies summarized ICU and non-ICU patients separately. Probably, most of patients in the rest of the studies did not appear to be transferred to the ICU. Therefore, a sensitivity analysis was difficult to perform.

Minor concern, Line 216: COVID-19 survivors were multifarious with "an" incidence of 5-17%.

Thank you. We corrected this.

---

## [Decision Letter · Decision Letter 1]

12 Oct 2022

PONE-D-21-13798R1Incidence of long-term post-acute sequelae of SARS-CoV-2 infection related to pain and other symptoms:A systematic review and meta-analysisPLOS ONE

Dear Dr. Shiga,

Thank you for submitting your manuscript to PLOS ONE. After careful consideration, we feel that it has merit but does not fully meet PLOS ONE’s publication criteria as it currently stands. Therefore, we invite you to submit a revised version of the manuscript that addresses the points raised during the review process.

We look forward to receiving your revised manuscript.

Kind regards,

Miquel Vall-llosera Camps

Senior Editor

PLOS ONE

Journal Requirements:

Additional Editor Comments:

I would like to sincerely apologise for the delay you have incurred with your submission. As mentioned previously, during our final internal checks on this submission, we noticed a potential concern regarding the quality of the peer review process. To ensure that your work received a thorough and objective evaluation at PLOS ONE, we considered necessary to invite additional reviewers. We have now received their completed reviews; the comments are available below.

Although the previous reviewers are happy with your response to their previous comments, the additional reviewers some raised scientific concerns about your study. In particular, reviewer#5 raised concerns that the literature search of your manuscript is out of date. We acknowledge that this concern has been raised previously during this peer review and that you responded that due to the high volume of publications rewriting the manuscript would not be possible. In consideration of your response, to address this concern we would request including in your manuscript a discussion on research published after the search was complete and discussion of your study in the context of the recent published literature.

Please revise the manuscript to address reviewer#5 and #6 comments in a point-by-point response. Please note that the revised manuscript might need to undergo further review, but if you can address these comments satisfactorily the manuscript will be ready to move forward.

Reviewers' comments:

Reviewer's Responses to Questions

**Comments to the Author**

1. If the authors have adequately addressed your comments raised in a previous round of review and you feel that this manuscript is now acceptable for publication, you may indicate that here to bypass the “Comments to the Author” section, enter your conflict of interest statement in the “Confidential to Editor” section, and submit your "Accept" recommendation.

Reviewer #1: All comments have been addressed

Reviewer #3: All comments have been addressed

Reviewer #4: All comments have been addressed

Reviewer #5: (No Response)

Reviewer #6: All comments have been addressed

2. Is the manuscript technically sound, and do the data support the conclusions?

Reviewer #1: Yes

Reviewer #3: Yes

Reviewer #4: Yes

Reviewer #5: No

Reviewer #6: Yes

3. Has the statistical analysis been performed appropriately and rigorously? 

Reviewer #1: Yes

Reviewer #3: Yes

Reviewer #4: Yes

Reviewer #5: Yes

Reviewer #6: Yes

4. Have the authors made all data underlying the findings in their manuscript fully available?

Reviewer #1: Yes

Reviewer #3: Yes

Reviewer #4: Yes

Reviewer #5: Yes

Reviewer #6: Yes

5. Is the manuscript presented in an intelligible fashion and written in standard English?

Reviewer #1: Yes

Reviewer #3: Yes

Reviewer #4: Yes

Reviewer #5: Yes

Reviewer #6: Yes

6. Review Comments to the Author

Reviewer #1: The authors addressed reviewer's comment and revised the manuscript accordingly. I suggest the current study accepted for publication.

Reviewer #3: The authors submitted a revised version of the article along with thorough explanation of the way by which the changes and corrections were made. I have no serious flaws to the article in its revised version.

Reviewer #4: The authors submitted a revised version of the paper along with the clear explanation if the ways by which the corrections were made. I have no serious flaws to the article in its revised version.

Reviewer #5: This meta analysis is totally out of date. The last search was January 2021, now we are September 2022. One year in COVID-19 research implies that several papers and MA about post-COVID pain have been published. This paper does not add nothing to the literature and does not include most updated data.

Reviewer #6: This manuscript describes a systematic review and meta-analysis to estimate the incidence rate for pain and other symptoms due to “long COVID.”

This study used several large electronic databases (including MediRxiv and BioRxiv) to search articles before Jan 15, 2021. In total, 1290 articles were identified in the literature search, and finally, 35 studies were included in the study after a rigorous screening and selection process. The analyses followed a PRISMA standard for meta-analysis, and the inter-study heterogeneity was assessed using I^2 statistics. In addition, the authors investigated the potential source of heterogeneity using meta-regression with a mixed-effects model. Furthermore, the publication bias was assessed by Egger’s P statistics and funnel plots.

Overall, the study method is solid and rigorous. This study provides comprehensive results for long COVID related pain and other symptoms.

I have two questions just for curiosity.

1. The result “every one month of follow-up corresponds to an increase of 1.45 units (45% increase) in prevalence in patients who developed neuralgia after acute COVID-19 infection” indicate the longer follow-up time is associated high incidence rate of neuralgia. Is that because neuralgia will take time to develop, so the studies with short follow-up time can not detect the development of neuralgia?

2. Will different regions of study (e.g. Europe, US and China) be a potential factor contributing to the heterogeneity, as people from different regions may have different tolerances or thresholds of pain?

7. PLOS authors have the option to publish the peer review history of their article (what does this mean?). If published, this will include your full peer review and any attached files.

Reviewer #1: No

Reviewer #3: **Yes: **Alexander E Berezin, FESC, Professor of Medicine, MD, PhD, DSci

Reviewer #4: No

Reviewer #5: No

Reviewer #6: No

---

## [Author Response · Author response to Decision Letter 1]

13 Nov 2022

The original reviewers’ comments are expressed in “bold” type, and our comments are expressed in “plain” type.

Reviewer #1: The authors addressed reviewer's comment and revised the manuscript accordingly. I suggest the current study accepted for publication.

Thank you very much for your comment and recommendation.

Reviewer #3: The authors submitted a revised version of the article along with thorough explanation of the way by which the changes and corrections were made. I have no serious flaws to the article in its revised version.

Thank you for your comments.

Reviewer #4: The authors submitted a revised version of the paper along with the clear explanation if the ways by which the corrections were made. I have no serious flaws to the article in its revised version.

Thank you for your comments.

Reviewer #5: This meta analysis is totally out of date. The last search was January 2021, now we are September 2022. One year in COVID-19 research implies that several papers and MA about post-COVID pain have been published. This paper does not add nothing to the literature and does not include most updated data.

We totally agree with your comments.

To be honest, this paper has a long history. It was originally published in MedRxiv on April 8, 2021, before its submission to PLOS One. The manuscript revision process took more than six months, after which the manuscript was officially accepted by PLOS One on August 13, 2022.

One month later, after our payment to the journal, however, acceptance was suddenly retracted by the editorial office due to a potential concern regarding the quality of the peer review process. So, this explains the delay.

During this time, thousands of articles on long COVID were published, so we think that would be impossible, within this short period of revision time, to screen and extract new data, re-analyze it, and extensively rewrite the manuscript. We believe that the previous reviewers were happy with our response to these circumstances. We kindly ask for your understanding in advance.

However, as the editor requested, we attempted to discuss our results in the context of the recently published literature. We found one similar meta-analysis published by Fernandez-de-las-Penas et al. in 2022 (This paper was submitted on 26 May, 2021, when our preprint has already published; therefore, we were first). In the paper, the most frequent pain-related symptoms and their incidences were comparable with those in our studies. For instance, chest pain is the most reported, and its incidence ranges between 7.8-23.6%. We added this in the Discussion section as follows: “During the course of our study, a similar meta-analysis related with pain were published [63]. Although this might weaken the originality of our study, in the publication, the most frequently reported pain-related symptoms and their incidence were comparable with those reported in our studies. For instance, chest pain is the most reported symptoms, and its incidence ranges between 7.8-23.6%.

Reviewer #6: This manuscript describes a systematic review and meta-analysis to estimate the incidence rate for pain and other symptoms due to “long COVID.”

This study used several large electronic databases (including MediRxiv and BioRxiv) to search articles before Jan 15, 2021. In total, 1290 articles were identified in the literature search, and finally, 35 studies were included in the study after a rigorous screening and selection process. The analyses followed a PRISMA standard for meta-analysis, and the inter-study heterogeneity was assessed using I^2 statistics. In addition, the authors investigated the potential source of heterogeneity using meta-regression with a mixed-effects model. Furthermore, the publication bias was assessed by Egger’s P statistics and funnel plots.

Overall, the study method is solid and rigorous. This study provides comprehensive results for long COVID related pain and other symptoms.

I have two questions just for curiosity.

1. The result “every one month of follow-up corresponds to an increase of 1.45 units (45% increase) in prevalence in patients who developed neuralgia after acute COVID-19 infection” indicate the longer follow-up time is associated high incidence rate of neuralgia. Is that because neuralgia will take time to develop, so the studies with short follow-up time can not detect the development of neuralgia?

First of all, we know you are busy, and we appreciate you taking the time to write your comments.

Regarding neuralgia, there are only three studies with high heterogeneity. Because of the small amount of available information, definite conclusions cannot be drawn. This statistic (45% increase per 1.45 units) is just an estimation calculated from the limited sample data available. We have already addressed this previously in the Discussion section as follows: “In any case, we are aware that these statistical models are preliminary and exploratory, and 53% of symptoms were not explainable despite three typical covariates being incorporated into the model. Symptoms of long COVID are reported to be on-and-off, cyclic or multiphasic [5], which is why the linear regression model did not fit well.”

2. Will different regions of study (e.g. Europe, US and China) be a potential factor contributing to the heterogeneity, as people from different regions may have different tolerances or thresholds of pain?

We think it’s probably yes. According to the meta-analysis published by Alkodyami et al. in 2022, lower incidences of fatigue, dyspnea, and loss of smell and taste were reported in Asian populations. However, meta-regression analysis with region as the covariate is beyond the aim of our study and was not included in our initial protocol, so we added this as a limitation in the Discussion section as follows: “Lastly, studies performed in different geographical regions might be a potential factor contributing to the heterogeneity, which was suggested in a recent meta-analysis [64].”

---

## [Decision Letter · Decision Letter 2]

27 Feb 2023

Incidence of long-term post-acute sequelae of SARS-CoV-2 infection related to pain and other symptoms:A systematic review and meta-analysis

PONE-D-21-13798R2

Dear Dr. Shiga,

We’re pleased to inform you that your manuscript has been judged scientifically suitable for publication and will be formally accepted for publication once it meets all outstanding technical requirements.

Kind regards,

Huzaifa Ahmad Cheema

Academic Editor

PLOS ONE

Additional Editor Comments (optional):

Due to the long delay associated with this manuscript, I believe it should be expedited for publication as soon as possible. I believe the comments of the authors have been suitably addressed and their are no major concerns.

Reviewers' comments:

Reviewer's Responses to Questions

**Comments to the Author**

1. If the authors have adequately addressed your comments raised in a previous round of review and you feel that this manuscript is now acceptable for publication, you may indicate that here to bypass the “Comments to the Author” section, enter your conflict of interest statement in the “Confidential to Editor” section, and submit your "Accept" recommendation.

Reviewer #1: All comments have been addressed

Reviewer #3: All comments have been addressed

Reviewer #5: (No Response)

Reviewer #6: All comments have been addressed

2. Is the manuscript technically sound, and do the data support the conclusions?

Reviewer #1: Yes

Reviewer #3: Yes

Reviewer #5: No

Reviewer #6: Yes

3. Has the statistical analysis been performed appropriately and rigorously? 

Reviewer #1: Yes

Reviewer #3: Yes

Reviewer #5: No

Reviewer #6: Yes

4. Have the authors made all data underlying the findings in their manuscript fully available?

Reviewer #1: Yes

Reviewer #3: Yes

Reviewer #5: No

Reviewer #6: Yes

5. Is the manuscript presented in an intelligible fashion and written in standard English?

Reviewer #1: Yes

Reviewer #3: Yes

Reviewer #5: Yes

Reviewer #6: Yes

6. Review Comments to the Author

Reviewer #1: The revised manuscript addressed reviewers' and editor's suggestion adequately. I suggested this manuscript accepted for publication.

Reviewer #3: The authors gave comprehensive responces to reviewers' comments and deeply changed the manuscript. I am satisfied about the revised version of the paper.

Reviewer #5: this reviewer understand the situation that the authors comment about their manuscript but publishing a meta-analysis conducted in 2021 in 2023 is not feasible. I apologize, my opinion is the same, authors must update the review and focus only on pain symptoms.

Reviewer #6: (No Response)

7. PLOS authors have the option to publish the peer review history of their article (what does this mean?). If published, this will include your full peer review and any attached files.

Reviewer #1: **Yes: **Wei-Chih Chen

Reviewer #3: No

Reviewer #5: No

Reviewer #6: No

---

## [Editor Report · Acceptance letter]

12 Aug 2022

PONE-D-21-13798R1 

Incidence of long-term post-acute sequelae of SARS-CoV-2 infection related to pain and other symptoms:A systematic review and meta-analysis 

Dear Dr. Shiga:

I'm pleased to inform you that your manuscript has been deemed suitable for publication in PLOS ONE. Congratulations! Your manuscript is now with our production department. 

Kind regards, 

on behalf of

Dr. Robert Jeenchen Chen 

Academic Editor

PLOS ONE